# ATENA: A Web-Based Tool for Modelling Metal Oxide Nanoparticles Based on NanoFingerprint Quantitative Structure–Activity Relationships

**DOI:** 10.3390/molecules29102235

**Published:** 2024-05-10

**Authors:** Francesc Serratosa

**Affiliations:** Computer Science and Math Department, Universitat Rovira i Virgili, 43007 Tarragona, Catalonia, Spain; francesc.serratosa@urv.cat

**Keywords:** metal oxide nanocompound, chemical 3D structure, NanoFingerprint, graph embedding, graph regression

## Abstract

Modelling size-realistic nanomaterials to analyse some of their properties, such as toxicity, solubility, or electronic structure, is a current challenge in computational and theoretical chemistry. The representation of the all-atom three-dimensional structure of a nanocompound would be ideal, as it could account explicitly for structural effects. However, the use of the whole structure is tedious due to the high data management and the structural complexity that accompanies the surface of the nanoparticle. Developing appropriate tools that enable a quantitative analysis of the structure, as well as the selection of regions of interest such as the core-shell, is a crucial step toward enabling the efficient analysis and processing of model nanostructures. The aim of this study was twofold. First, we defined the NanoFingerprint, which is a representation of a nanocompound in the form of a vector based on its 3D structure. The local relationship between atoms, i.e., their coordination within successive layers of neighbours, allows the characterisation of the local structure through the atom connectivity, maintaining the information of the three-dimensional structure but increasing the management ability. Second, we present a web server, called ATENA, to generate NanoFingerprints and other tools based on the 3D structure of the nanocompounds. A case study is reported to show the validity of our new fingerprint tool and the usefulness of our server. The scientific community and also private companies have a new tool based on a public web server for exploring the toxicity of nanocompounds.

## 1. Introduction

Binary metal oxide nanoparticles are chemical compounds composed of two types of atoms, oxygen and any metal, for instance, Al, Cu, Fe, Ti or Zn. The most common metal oxide nanoparticles are Al_2_O_3_, CuO, Fe_2_O_3_, TiO_2_, SiO_2_ and ZnO. The analysis and prediction of their reactivity and toxicity are crucial for better understanding these compounds and their use in the industry [1].

Given that the experimental evaluation of the safety of chemicals is expensive and time consuming, computational methods have been found to be efficient alternatives for predicting models based on quantitative structure–activity relationships (QSARs) [2,3,4,5,6]. These alternatives are based on the assumption that the activity of a substance is related to its structure for the potential toxicity prediction and environmental impact of new nanomaterials before mass production. This structural analysis, which deduces some properties of the compounds, can be the analysis of the three-dimensional structure. The recent appearance of crystallographic tools for the generation of three-dimensional structures is remarkable [7]. Nevertheless, the management of the whole structure, in terms of the size of the nanoparticle, is tedious in computational chemistry, due to the management of a large amount of data, as the number of atoms increases exponentially with the size [8]. This is the reason why some methods discard the core of the nanocompound and only keep the three-dimensional structure of the shell of the nanocompound. It has been demonstrated that the reactivity or toxicity of the nanoparticle mostly depends on the shell of the nanoparticle. This is the reason why some methods have been presented to deduce the thickness of the shell of the nanoparticle with the aim of discerning between core and shell and putting more effort on understanding the properties of the shell [9].

Several QSAR models have been presented to predict the toxicity of nano compounds. In our case, we are interested in metal oxide nanocompounds. In this specific domain, the size of a nanocompound has been demonstrated to be a crucial parameter for the toxicity prediction. We can find nanocompounds that are toxic in small sizes but not in larger ones, and vice versa. This is the reason why the presented models always have size as a parameter. Usually, these models are based on a linear regression. One of the oldest models for metal oxide nanocompound toxicity prediction was [5]. The model predicts the lactate dehydrogenase release (LDH) of TiO_2_ and ZnO. The parameters of the linear regression are concentration (mg/L), size in Pbs (nm) and size (nm). Considering another application, the authors of [4] predicted the toxicity of metal oxide in log(1/EC50). The parameters were the enthalpy of the formation of the metal oxide (Kcal/mol) and the Mullikens electronegativiy (eV). A cytotoxicity prediction was presented in [3]. In this case, the parameters of the linear equation were the size (nm), volume (nm^3^), period of the metal and the automatisation of the metal oxide (Kcal/eqv). Another model that also predicts the cytotoxicity was proposed in [2], which is based only on the information of qmelec and lzelehho in atomic units. Finally, the authors of [6] predicted the toxicity of metal oxide based on the size and hydrosize of the nanocompound and the surface charge (eV) and area (nm^2^), as well as other parameters.

The three-dimensional structures of nanoparticles can be represented as attributed graphs [10]. An attributed graph is a mathematical model of an object composed of two types of representations: nodes and edges. Nodes are individual components in the object, and edges are relationships between these components. Attributed graphs have been applied in machine learning for a long time in applications ranging from character recognition to social network exploring, among others, and more precisely, in Ligand-Based Virtual Screening [8,10,11]. In our case, atoms in the nanoparticle are represented by nodes, and chemical bounds, by edges. Moreover, nodes and edges can be attached with some attributes, which are the main information of these local parts of the objects and their relationships. The node attributes might be the type of atom (O, Al, Cu, Fe, Ti or Zn) or its energy, among other metrics. And the edge attribute might be the type of bond between atoms or some relationships between the nodes it connects.

The objective of this paper was twofold. On the one hand, we present a new fingerprint model for metal oxide nanoparticles called NanoFingerprint. It is based on i. deducing the three-dimensional structure of a nanocompound, ii. selecting only the atoms that are in the shell of the nanoparticle, iii. representing the shell as an attributed graph, and iv. filling the NanoFingerprint representation by extracting the appearance of some subgraphs in this graph. Note that a case study was conducted to discover its applicability. On the other hand, we present a web server tool, called ATENA, to generate NanoFingerprints and also to use toxicity QSAR models based on classical models and this new representation. The aim of ATENA is to be a public tool for the toxicity analysis of metal oxide nanocompounds and also for the analysis of their crystallographic structures.

NanoFingerprints were initially presented at a congress [12] without detailing examples, generation algorithms, their applicability or the web server. The aim of this paper is to explain to users (chemical labs or researches) this new model and how to use the web server.

## 2. Results

In this section, we show the usefulness of this representation in three different scenarios:Applications in which we want to predict the toxicity of nanocompounds.Applications in which we want to apply crystalline optimisation to nanocompound three-dimensional structures.Applications in which we want to generate NanoFingerprints that represent large nanocompounds to deduce their toxicity without the need of having their 3D structure.

### 2.1. Toxicity Prediction

Some QSAR models have been reported for the prediction of the toxicity of metal oxide nanocompounds [13]. In this section, we show the usability of NanoFingerprints for toxicity prediction by defining new QSARs. The main idea is to increase the data used to compute the prediction in other QSAR models, such as the ones in [5,6], and, in doing so, increase the accuracy of these predictions. Figure 1 exemplifies a general method used to predict the activity of a nanocompound based on a classical QSAR model (top) and our NanoFingerprint model (bottom). Note that we use properties extracted from the in vitro process and also properties extracted from the theoretical structure of the nanocompound. Rectangles represent processes or models, and circles represent data.

In 2015, a QSAR model for lactate dehydrogenase release (LDH) prediction based on five parameters, size, size in water, size in PBS, concentration and Zeta potential, was presented in [5]. Then, in 2021, a QSAR model for the toxicity assessment of metal oxide nanoparticles using nine physicochemical features, CoreSize, HydroSize, SurfCharge, SurfArea, Ec, Time, Dose, Eneg and NOxygen, was presented in [6]. These are two examples of QSAR models applied to metal oxide nanoparticles. In the first one, the QSAR is based on a linear regression, and in the second one, it is based on a logistic regression.

We used these models to test the validity of our QSAR because data and model parameters have different natures (the data extracted from [5,6] are detailed in the Appendix A). In both cases, we used the same dataset and also the same method to divide the dataset into learning and testing samples as that reported in the papers. In the first paper, there were only 22 samples used to predict the LDH of TiO_2_ and a *leave-one-out* training method was used. In the second one, there were 483 samples, and a *cross-validation partition* with 10 folders was used. Moreover, we increased the number of samples, as the original paper did to compensate for the data unbalancing problem. We performed three experiments on both datasets:Test1: Computing the regression with their parameters (five parameters in [5] and nine parameters in [6]).Test2: Computing the regression with sections 1 and 2 of the NanoFingerprint (maximum number of bonds = 5 and shell thickness = 5 A).Test3: Computing the regression with the concatenation of parameters in Test1 and Test2.

Table 1 and Table 2 show the main validation metrics applied to the data in [5,6]. We concluded the following from this validation:Structural information for TiO_2_ne is not enough for the toxicity prediction (the balanced accuracy, precision and recall were lower in Test2 than in Test1. Moreover, the false negatives were larger in Test2 than in Test1).Nevertheless, it helps to increase the quality of the toxicity prediction compared to global ones (Test3 returns better validation parameters than Test1).

### 2.2. Crystalline Optimisation of the Three-Dimensional Structure

The generation of the three-dimensional structure of a nanocompound is usually carried out in two steps. In the first step, a crystalline structure is generated through the replication of an initial core. And in the second step, this structure is slightly modified using an optimisation process based on moving the atoms such that the forces on each atom, and the total energy, are minimised. Clearly, the second step is computationally much more demanding than the first one. The bulk-cut structure contains many undercoordinated atoms in the shell, and their number will decrease after optimisation. The NanoFingerprint was used to track the number of undercoordinated atoms before and after optimisation in a TiO_2_ spherical nanoparticle with a 3 nm diameter to validate the methodology. The most frequent substructures were obtained, and they could be localised based on the output of the NanoFingerprint.

We extracted the NanoFingerprint of five three-dimensional structures of TiO_2_ with sizes of 1 nm, 1.5 nm, 1.6 nm, 1.7 nm and 3.2 nm, initially obtained with the tool [7]. Then, we optimised these three-dimensional structures with the VASP code [14,15], using the computational setting in [16] and we extracted their NanoFingerprints (maximum number of bonds per atom = 5 and shell thickness = 5 A). Finally, we subtracted the optimised NanoFingerprints from the non-optimised NanoFingerprints.

Figure 2 shows an illustration of the output of the NanoFingerprints for a 3.2 nm non-optimized nanoparticle. The coordination (number of first neighbours bonded to an atom) is defined as O[0,x] or Ti[x,0]; there are six singly coordinated Ti sites (marked as Ti[1,0]), 48 singly coordinated O sites (marked as O[0,1]), and 624 three-fold coordinated O sites (marked as O[0,3]). Moreover, the local environment is also captured by the combination of pairs of O and Ti sites; there are 24 singly coordinated O sites O[0,1] connected to octahedral Ti sites Ti[6,0], of which there are 24. Additionally, the atoms are labelled as core or shell if they are located within the radius specified in step 2, and an XYZ file is generated with this information.

The optimisation procedure leads to a decrease in the number of undercoordinated sites. For instance, we realised that there is a tendency of there being less O(1) and more O(4) or O(5) in the optimised structure of the TiO_2_ than in the original one, independently of the size. Note that the volume of the particle changes as the interaction between atoms is optimised. For this reason, we considered the atoms that belong to the shell in the optimised structure to be the ones that belong to the shell in the non-optimised structure. Thus, the number of atoms in that region remains the same.

### 2.3. Generative Model for NanoFingerprint Prediction

We aimed to artificially generate the NanoFingerprint of a specific nanocompound given its size. As previously commented, the generation of the three-dimensional structure is needed to deduce the NanoFingerprint, which is a computationally expensive task for medium to large nanocompounds. Thus, the general idea is to model a QSAR that incorporates NanoFingerprints (as described in the Toxicity prediction section), but the NanoFingerprint was artificially predicted by a machine learning model. To do so, the machine learning method, based on a generative model, generates the entire NanoFingerprint with the introduction of the type of metal, the size, the shell thickness and the maximum number of bonds.

In this way, we could extrapolate a NanoFingerprint of a huge nanocompound, which has never been obtained. Then, we could approximate some properties of this nanocompound (for instance, its toxicity) without the need of having it. Imagine that we deduce, in the lab, the toxicity of some small chemical nanocompounds but we cannot generate larger ones due to some practical reasons. In this case, we could approximate its NanoFingerprint and also predict its toxicity without generating or buying it.

Figure 3 shows the process of learning the QSAR based on a classical model (up) and based on a generative model (down). It is similar to Figure 1, but the NanoFingerprint generator has been substituted with the generative model. Note that, in this case, the generation of the three-dimensional structure is not needed, and instead, a generative model is used.

Several neural networks were trained as a generative model, but due to the low number of samples, over-training problems always appeared. For this reason, we moved to generate the NanoFingerprint with a logarithmic regression, and we achieved satisfactory results.

Table 3 shows the classification metrics considering four different architectures: (1) the data and method in [6]; (2) regression on the data in [6] concatenated to (represented by the symbol ‘+’) the embedding vector generated in [17]; (3) regression on the NanoFingerprint generated using the 3D structure of the data in [6]; and (5) regression on the NanoFingerprint. In (3) and (5), the NanoFingerprints were generated using the method reported in [12]. Moreover, the classification metrics for the NanoFingerprints appear in bold. Note that (4) is the same architecture as (3). Finally, (6) is the same architecture as (5).

We first realised that adding the NanoFingerprint embedding results in a higher quality in the classification process. In addition, using the NanoFingerprint generated by our regressor creates slight decreases in the classification metrics but is still the best option. Thus, we generated a method that is fast to generate new samples and maintains a high quality. The original data were composed of 483 different nanocompounds with different sizes and are available in Appendix A.

## 3. ATENA: A Web Server Tool

The web server ATENA (https://atena.urv.cat/model/ (accessed on 7 May 2024)) is devoted to analysing metal oxide nanocompounds from the structural point of view. This means that the 3D information of the compound is required. From an initial description of the nanocompound formatted as an XYZ file, some toxicity predictions can be performed. Moreover, some analysis can be performed, such as on the appearance of specific local patterns inside the nanocompound, which are described in XYZ or GRF format, and also on the appearance of combinations of atoms in the shell that are know to be toxic. XYZ and GRF formats are explained and exemplified on the website.

There are other web servers for the toxicity prediction of chemical compounds [18]. Nevertheless, these web servers propose general toxicity predictions and are not based on structural information (XYZ file) and or on nanocompounds.

### 3.1. Website Description

The website is composed of three main functions: *NanoFingerprint*, *Toxicity prediction* and *Subcomponent search*. There is also other material such as examples and general information. Note that this website is frequently updated with new models.

NanoFingerprintThis computes a NanoFingerprint given a metal oxide nanoparticle described in a .XYZ file. Two parameters are required: shell thickness and the maximum number of bonds per atom, which are the first two values of the NanoFingerprint. The website returns a .txt file that contains the NanoFingerprint. Nanocompounds in an XYZ format can be structures that have been optimised or not. There are some websites that provide these structures. For instance, *the crystallographic tool for the construction of nanoparticles* (https://nanocrystal.vi-seem.eu/ (accessed on 7 May 2024)) reported in [7].Toxicity predictionThis computes the toxicity of some nanocompounds with reported models, which are not based on NanoFingerprints. This section will increase from time to time when new models are incorporated. For instance, there is a model for predicting the cytotoxicity of TiO_2_ and ZnO nanoparticles using empirical descriptors [5]. The aim is to use these models for comparisons with models based on structural nanocompounds.Subcomponent searchThis returns the appearances of some specific local structures that could appear in the nanocompound. It is required to introduce the nanocompound in XYZ format and the local structure in XYZ or GRF format (described on the website). Moreover, it is also required to introduce the thickness of the shell (if the user wants to search the local structure in the whole compound, a large number can be introduced). No other web server has been found with this functionality.

### 3.2. Examples of Website Use

This shows an example of the use of our website with the three main functions.

ModellingFigure 2 shows an example of the analysis of an anatase nanoparticle TiO_2_ with a size of 3 nm. From this nanoparticle, we generated a NanoFingerprint with a shell thickness of 0.4 nm and a maximum number of bonds of 10. To do so, the first step of the algorithm implemented in the server is to deduce the shell atoms and bonds (Figure 2(left)) and then extract the NanoFingerprint (Figure 2(centre)). We highlighted some specific local structures located in the shell. The elements in section 1 of the NanoFingerprint are as follows: 4, 10, 30, 22, 414, 217. We show only some elements of the vector due to space restrictions. The total length is $6+2*10+2*10^{2}+3*10^{4}$ = 30,226. Finally, part of the XYZ file that was introduced and has the information of the an anatase nanoparticle TiO_2_ (3 nm) is shown in Figure 2(right).Toxicity predictionThe supplementary data of ref. [5], which describes a model for predicting the lactate dehydrogenase release (LDH) of a TiO_2_, include two compounds of TiO_2_ with a 30 nm diameter and a concentration of 100 mg/L that had LDH releases of 1.04 and 1.09, respectively. We introduced these parameters (30 nm and 100 mg/L) into our website, and our model returned 1.02 LDH. This result has a relative error of 1.9% and 6.4%.Subcomponent searchFigure 4 shows a TiO_2_ compound with a size of 2 nm, and Figure 5(right) shows TiO_2_ a compound of 6 Angstrom. Our aim was to search the appearances of the smaller compound in the shell of the larger one. We imposed a thickness shell of 4 Angstrom, and Figure 5(left) shows the output generated by our website. It seems that there are four appearances of the smaller compound in the shell of the larger one. Note that the output of the server is not the image but an XYZ file. These images were generated using the Matlab function molviewer, although there are other packages for these visualisations.

## 4. Discussion

NanoFingerprints were tested using ATENA. Three different types of applications were tested: toxicity prediction, crystalline optimisation and NanoFingerprint artificial generation. In the first case, we arrived at the conclusion that it is worth it to concatenate NanoFingerprints with global properties since this combination achieves better results than other known models. Considering crystalline optimisation, NanoFingerprints also achieved good results since they keep the information of substructures that are known to have some properties, such as toxicity. Finally, artificially generated NanoFingerprints were used to deduce the toxicity level of nanocompounds, which were too big to for the 3D structure obtained. The experiments provided competitive results with respect to NanoFingerprints generated using the classical algorithm (which is quadratic with respect to the number of atoms).

All in all, the practical experiments showed that ATENA is a new and interesting tool for the exploration of toxicity in metal oxide nanocompounds.

## 5. Materials and Methods

This section is composed of three subsections covering the global parameters needed to generate a NanoFingerprint, some basic definitions and the NanoFingerprint definition.

### 5.1. NanoFingerprint Generative Parameters

Three parameters are needed to generate NanoFingerprints (Figure 6):Shell thickness: A positive real number that defines the external radius of the nanocompound such that the atoms in the defined volume are considered to be inside the shell, and thus, these atoms influence the generation of the NanoFingerprint.Maximum bonds: Natural numbers that define the maximum number of bonds per atom that we consider to generate the NanoFingerprint. A larger number makes a larger NanoFingerprint and a larger chance of having more null values in the fingerprint.3D structure: A file in an XYZ format that contains the 3D structural information of the NanoFingerprint. Note that it could be considered to introduce an “emptied” 3D structure, an XYZ file that only contains the atoms in the shell. In this case, the XYZ is smaller, and the generation of the NanoFingerprint is faster.

### 5.2. Basic Definitions

A *local structure* is a small set of close atoms and their bonds in the nanocompound. In this section, we present some *local structures* that are used to define, in an organised manner, a NanoFingerprint as follows:

O(x): This represents an oxygen that has a covalent bond to another *x* oxygen or metal, independently.

M(x): Similarly, this represents a metal that has a covalent bond to another *x* oxygen or metal, independently.

O(x,y): A local structure composed of a central oxygen connected to *x* oxygens and *y* metals. Note that these *x* oxygens and *y* metals could be connected to other atoms.

M(x,y): Similarly, a local structure composed of a central metal connected to *x* oxygens and *y* metals. Again, note that these *x* oxygens and *y* metals could be connected to other atoms.

O(x,y)-$O(x^{\prime },y^{\prime })$: A structure that is composed of two of the previous ones. It is composed of an O(x,y) and an $O(x^{\prime },y^{\prime })$ with a central oxygen that are connected by a bond.

M(x,y)-$M(x^{\prime },y^{\prime })$: In a similar way, this is composed of an M(x,y) and an $M(x^{\prime },y^{\prime })$ with a central metal that are connected by a bond.

O(x,y)-$M(x^{\prime },y^{\prime })$: Finally, this is a similar structure but it connects an O(x,y) and an $M(x^{\prime },y^{\prime })$ with central atoms that are connected.

### 5.3. NanoFingerprint Definition

A NanoFingerprint is a vector of numbers that counts the number of appearances of *local structures* in the shell of the nanocompound. All values are natural numbers except for the third element, which is a positive real number. It is split up in four main sections: global information, atomic information, bond information and structural information. Note that the number of atoms and bonds involved in each local structure increases in each section; that is, larger local structures are considered.

Section 1: Global informationThe first section accounts for the global information of the structure and the two parameters used to generate the NanoFingerprint. It is composed of 6 values:1: Shell thickness in Angstroms: This is the first algorithm input parameter. Only the most external atoms that are in the volume defined between the maximum radius and the maximum radius minus this parameter are considered. This parameter is imposed because it was shown that the atoms and bonds in the core of the nancompound have little or any influence on its reactivity or toxicity [9]. In the case that all atoms of the compounds are to be considered, this parameter has to be set as a value larger or equal than half the size of the nanocompound (the size is the third value in the NanoFingerprint).2: Maximum number of bonds per atom: This is the second algorithm input parameter. Atoms with a larger number of bonds are not considered. It is assumed that they do not exist; thus, it is the responsibility of the user to impose a value such that any (or few) atoms are discarded. This parameter is needed to generate a fixed representation of the NanoFingerprint, independently of the local structure of the atoms. In the rest of the paper, this parameter is called MAX.3: Size in Angstrom: This is the maximum distance between two external atoms of the nanocompound. If the compound is spherical, it is the diameter.4: Atomic number of the metal: NanoFingerprints are thought to embed the structure of metal oxide nanocompounds with only one type of metal.5: Number of oxygen atoms in the shell.6: Number of metal atoms in the shell.Section 2: Atomic informationThis section is composed of 2∗MAX values, where MAX is the maximum number of bonds per atom (the second value of the first section). The first MAX atoms referred to the oxygen, and the other ones, to the metal.1: Number of O(1)…MAX: Number of O(MAX)MAX+1: Number of M(1)…2MAX: Number of M(MAX)Section 3: Bond informationThis section includes the information of the local structures O(x,y) and M(x,y).It is composed of $2{\left(MAX+1\right)}^{2}$ values.1: Number of O(0,0)…MAX+1: Number of O(0,MAX)MAX+2: Number of O(1,0)…2(MAX+1): Number of O(1,MAX)…${\left(MAX+1\right)}^{2}$: Number of O(MAX,MAX)${\left(MAX+1\right)}^{2}+1$: Number of M(0,0)…${\left(MAX+1\right)}^{2}+MAX+1$: Number of M(0,MAX)${\left(MAX+1\right)}^{2}+MAX+2$: Number of M(1,2)…${\left(MAX+1\right)}^{2}+2\left(MAX+1\right)$: Number of M(1,MAX)…${\left(MAX+1\right)}^{2}+{\left(MAX+1\right)}^{2}$: Number of M(MAX,MAX)Section 4: Structural informationThis section includes the information of the local structures O(x,y)-$O(x^{\prime },y^{\prime })$, M(x,y)-$M(x^{\prime },y^{\prime })$ and O(x,y)-$M(x^{\prime },y^{\prime })$. It is composed of $3{\left(MAX+1\right)}^{4}$ values.1. Number of O(0,0)-O(0,0)…${\left(MAX+1\right)}^{2}$: Number of O(0,MAX)-O(0,MAX)${\left(MAX+1\right)}^{2}+1$: Number of O(1,0)-O(1,0)…${\left(MAX+1\right)}^{4}$: Number of O(MAX,MAX)-O(MAX,MAX)${\left(MAX+1\right)}^{4}+1$: Number of M(0,0)-M(0,0)…${\left(MAX+1\right)}^{4}+{\left(MAX+1\right)}^{4}$: Number of M(MAX,MAX)-M(MAX,MAX)${\left(MAX+1\right)}^{4}+{\left(MAX+1\right)}^{4}+1$: Number of O(0,0)-M(0,0)${\left(MAX+1\right)}^{4}+{\left(MAX+1\right)}^{4}+{\left(MAX+1\right)}^{4}$: N. of O(MAX,MAX)-O(MAX,MAX)

The length of the NanoFingerprint is $6+2MAX+2{\left(MAX+1\right)}^{2}+3{\left(MAX+1\right)}^{4}$.

### 5.4. NanoFingerprint Example

Figure 7a shows the chemical nanocompound Al_2_O_3_ that has a size of 13 Angstroms. It is only composed of 30 atoms; 18 of them are oxygen (in red) and 12 of them are aluminium (in grey). We can draw this compound since we have its three-dimensional structure (The usual file format used to describe the three-dimensional structures of nanocompounds is ‘.XYZ’. It is a well-know and simple format that, for each atom, keeps the information of the type of atom and its position (x,y,z). No information about the bonds is included, and thus, it has to be deduced through chemical–physical properties). Moreover, Figure 7b shows its NanoFingerprint. The maximum number of bonds was 10 (there are not atoms with more bonds in this compound).

## 6. Conclusions

This paper presented a new model for describing metal oxide nanocompounds, which we have called NanoFingerprint. It is defined as a vector that contains information about the number of appearances of some specific local 3D structures in each cell. The computational cost of generating this QSAR given the 3D structure of the nanocompound is quadratic with respect to the number of atoms. Since the runtime could be non-admissible in some applications, we proposed a regression model that generates an approximation of a NanoFigerprint. Both the exact NanoFingerprints and the approximated ones were tested in a toxicity application and showed good performances compared to other known methods.

Moreover, a web server, called ATENA, was presented that allows users to generate a NanoFingerprint, analyse some toxicity models and search local structures in larger three-dimensional structures. This search could be used to detect possible combinations of atoms that generate toxicity.

We believe that this new QSAR model and the web server ATENA will be useful for the safe use of metal oxide nanocompounds and for the research on these materials, as we have seen a significant increase in their demand.

## Figures and Tables

**Figure 1 molecules-29-02235-f001:**
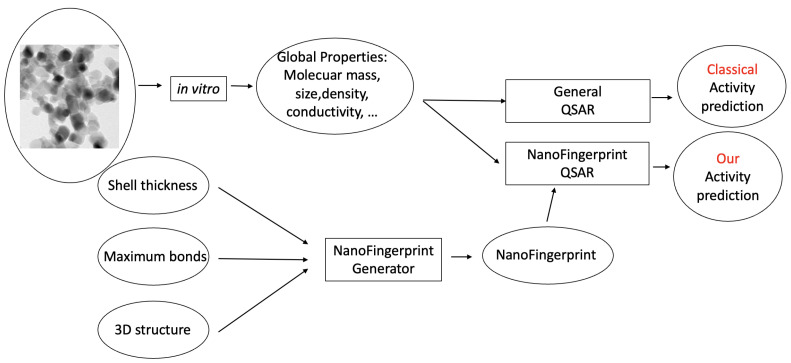
Two learning processes for predicting the activity of a nanocompound (TiO_2_ of 25 nm, as an example). The classical QSAR process is on top, and our NanoFingerprint process, is at the bottom. Our method increases the properties extracted in vitro with the properties extracted from the structure in silico.

**Figure 2 molecules-29-02235-f002:**
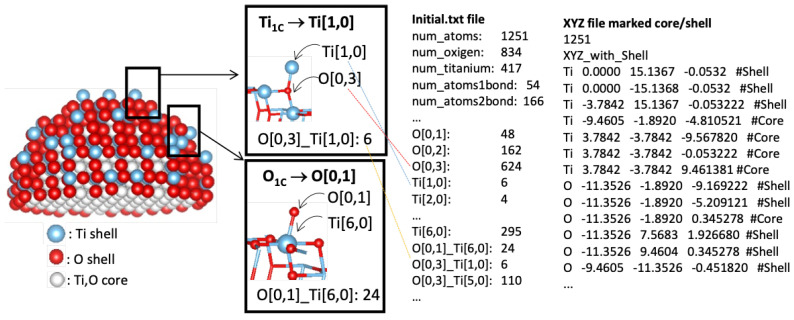
Schematic view of a TiO_2_ nanoparticle with a size of 3 nm. Atoms are classified as core or shell. Selected atoms with different chemical environments are indicated, with the corresponding generated NanoFingerprint. There is also part of the XYZ file introduced to the site.

**Figure 3 molecules-29-02235-f003:**
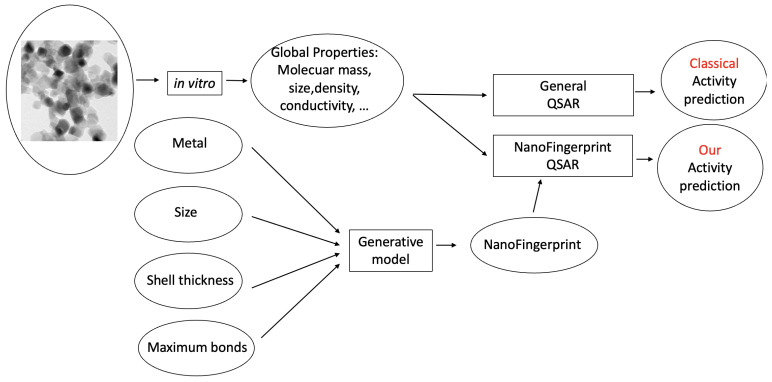
Two learning processes used to predict the activity of a nanocompound, similarly to Figure 1. Nevertheless, in this case, the NanoFigerprint is not generated by a three-dimensional structure but predicted by a generative model.

**Figure 4 molecules-29-02235-f004:**
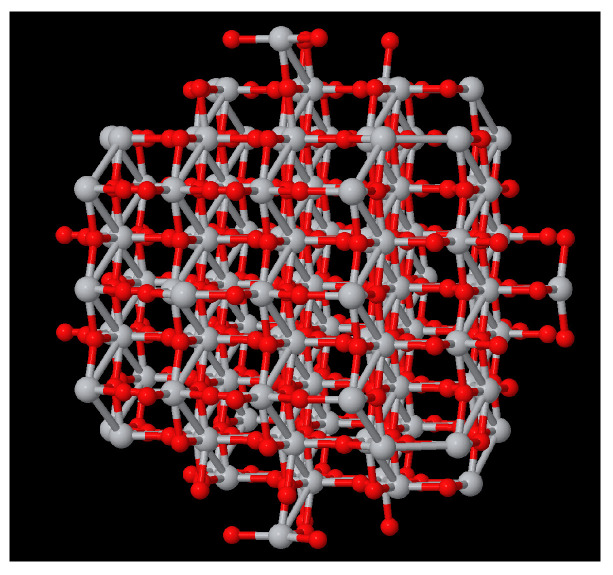
Visualisation of TiO_2_ of 2 nm.

**Figure 5 molecules-29-02235-f005:**
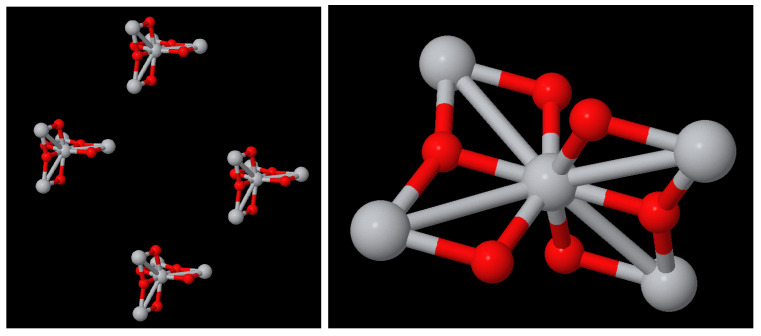
(**left**) Visualisation of the four appearances of TiO_2_ of 06A in the shell (thickness 4A) of TiO_2_ of 2 nm (Figure 4). (**right**) TiO_2_ of 06A.

**Figure 6 molecules-29-02235-f006:**
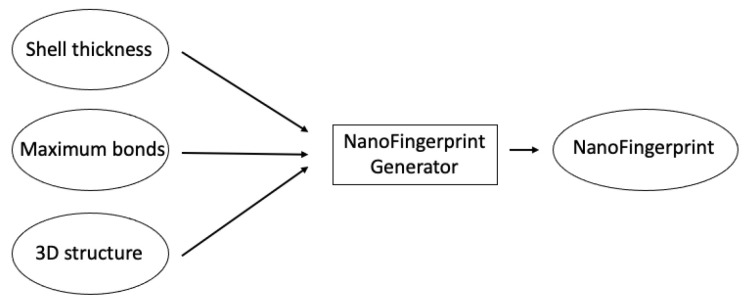
Main scheme followed to generate a NanoFingerprint.

**Figure 7 molecules-29-02235-f007:**
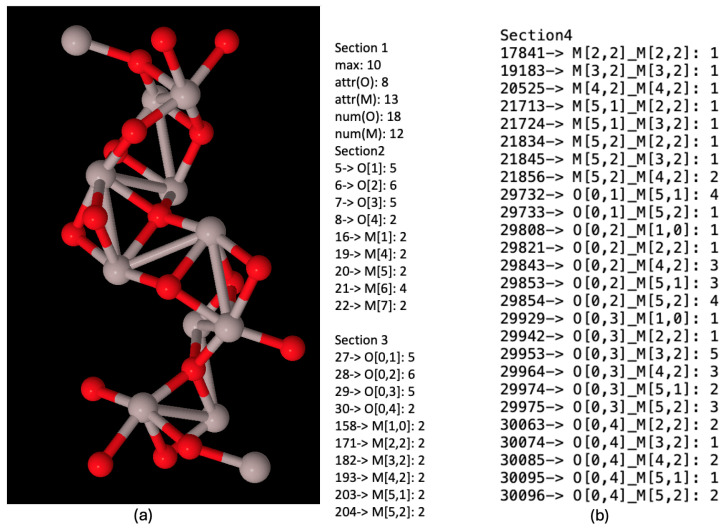
(**a**) Chemical nanocompound Al_2_O_3_ with a size of 13 Angstroms. Oxygen atoms in red and aluminium atoms in grey. (**b**) NanoFingerprint of chemical nanocompound Al_2_O_3_ with a size of 13 Angstroms. The first element of the NanoFingerprint concerns the maximum number of edges (10). The next two elements of the GraphFingerprint concern the elements being oxygen (atomic number: 8) and aluminium (atomic number: 13). Only the positions of the vector with a non-null value were selected. The positions appear at the beginning of each line.

**Table 1 molecules-29-02235-t001:** Means of main validation metrics concerning data in [6].

	Balanced Accuracy	Precision	Recall	False Positive	False Negative
Test1	0.81	0.81	0.98	0.8	13.9
Test2	0.77	0.77	0.98	0.8	18
Test3	0.83	0.82	0.98	0.8	12.8

**Table 2 molecules-29-02235-t002:** Mean square errors and standard deviations of data in [5].

	MSE	STD
Test1	0.12	0.08
Test2	0.26	0.16
Test3	0.12	0.08

**Table 3 molecules-29-02235-t003:** (1) Data and method in [6]; (2) regression on the data in [6] concatenated to (represented by the symbol ‘+’) the embedding vector generated in [17]; (3) regression on the GraphFingerprint generated using the 3D structure of the data in [6]; (4) is the same architecture as (3) but with the Fingerprint generated using our new method; (5) regression on the Fingerprint; and (6) is the same architecture as (5) but with the Fingerprint generated using our new method.

	Accuracy	Precision	Recall
(1) [6]	0.81	0.81	0.98
(2) [6] + GCN	0.56	0.80	0.88
(3) [6] + GraphFingerprint	0.93	0.92	0.98
(4) [6] + Generated NanoFingerprint	0.89	0.85	0.88
(5) Regression on GraphFingerprint	0.77	0.77	0.78
(6) Regression on Generated GraphFingerprint	0.75	0.76	0.71

## Data Availability

The data include the following tables in the Appendix A: 1. Toxicity prediction (Appendix A); 2. NanoFingerprint generation (NanoFingerprints can be generated at https://atena.urv.cat/model/ (accessed on 7 May 2024). Moreover, there is a tab on this website that includes some examples).

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
