# Peer review of "ATENA: A Web-Based Tool for Modelling Metal Oxide Nanoparticles Based on NanoFingerprint Quantitative Structure–Activity Relationships"

_molecules, 2024, doi:10.3390/molecules29102235_

Round 1

Reviewer 1 Report

Comments and Suggestions for Authors

The manuscript (molecules-2993774) presented a web server tool to generate Structural NanoFingerprints, the topic is interesting. To make it more publishable, the following comments should be addressed.

Comment 1: The writing of this manuscript should be carefully modified. For example, in the abstract, the author should clearly tell the reader the important finding and the novelty of this manuscript rather than just describe the aim and purpose.

Comment 2: The introduction part should be enriched to make more introduction of the traditional reported results to stress the significance of this work.

Comment 3: There are so many spelling and grammar mistakes, for example: "considered to" in Line 29 should be "is considered to be", the format of metal oxide should be unified, "is" in line 22 should be changed to "are". and many other places. Please carefully check the whole manuscript and make modifications.

Comment 4: For the description of the Figues, it is suggested to use Fig. 1 a, b rather thant Fig. (left) (right).

Comment 5:  In paragraph 1 of 2.3 it is mentioned that the third part is "Linking information", but in the following description the third part becomes "Bond information".

Comment 6: Please comment on the suitable application field of the new methodology. Catalyst modeling? 

Comment 7: It is strange have Figure in the conclusion part, please move the Figure to the main manuscript.

Comment 8: The references should be enriched. The following manuscripts are suggested for the citation: Molecular Catalysis 553 (2024) 113768, Materials Horizons, 2024 11, 2032-2040 

Comments on the Quality of English Language

Minor engish modification should be carried out.

Author Response

The manuscript (molecules-2993774) presented a web server tool to generate NanoFingerprints, the topic is interesting. To make it more publishable, the following comments should be addressed.

Comment 1: The writing of this manuscript should be carefully modified. For example, in the abstract, the author should clearly tell the reader the important finding and the novelty of this manuscript rather than just describe the aim and purpose.

Thank you for your comment. We have carefully read the paper and we have incorporated some new comments on the line you suggested. For instance, currently, the abstract finishes with the sentence “The scientific community and also private companies have a new tool to explore the toxicity of nanocompounds based on a public web server.”

Comment 2: The introduction part should be enriched to make more introduction of the traditional reported results to stress the significance of this work.

We agree that a new paragraph is needed presenting classical models. In fact, some classical models were compared in the experimental section but not properly commented in the introduction. We have added a whole paragraph presenting 5 classical QSAR models for nanocompund metal-oxide toxicity prediction. The ones that the 3D structure was possible to be generated are later compared in the experimental section.

Comment 3: There are so many spelling and grammar mistakes, for example: "considered to" in Line 29 should be "is considered to be", the format of metal oxide should be unified, "is" in line 22 should be changed to "are". and many other places. Please carefully check the whole manuscript and make modifications.

We have checked twice the paper to detect spelling mistakes. Thank you.

Comment 4: For the description of the Figures, it is suggested to use Fig. 1 a, b rather thant Fig. (left) (right).

 Done. Thanks.

Comment 5:  In paragraph 1 of 2.3 it is mentioned that the third part is "Linking information", but in the following description the third part becomes "Bond information".

 Done. Thanks.

Comment 6: Please comment on the suitable application field of the new methodology. Catalyst modeling? 

We have added the following sentence at the end of the introduction and before presenting the aim of the paper: “The aim of ATENA is to have a public tool for the toxicity analysis of metal-oxide nanocompounds and also for the analysis of their crystallographic structure.”. Moreover, a new Discussion section have been added, in which results and their applications are commented.

Comment 7: It is strange to have Figure in the conclusion part, please move the Figure to the main manuscript.

 Done. Thanks.

Comment 8: The references should be enriched. The following manuscripts are suggested for the citation: Molecular Catalysis 553 (2024) 113768, Materials Horizons, 2024 11, 2032-2040 

Three new references have been added that are cited in the introduction section. We have also added the suggested refrerences. Thanks.

Reviewer 2 Report

Comments and Suggestions for Authors

The article is devoted to the development of technologies that allow quantitative analysis of the structure, as well as the selection of areas of interest, to ensure effective analysis and processing of model nanostructures, as well as the ATENA web server for creating structural nanofingerprints. Today, the creation and improvement of tools that allow qualitative and quantitative analysis of various nanostructures is a very relevant area, especially with the use of digital fingerprints or nanofingerprints, as in this case. The article may be quite interesting for specialists working in the field of theoretical or experimental chemistry, however, I have several comments:

- the introduction is written too briefly, it lacks an analysis of works published on this topic or existing programs or services for creating models of nanostructures.

- in the introduction it is necessary to add information about the developments or scientific achievements previously obtained by the authors on this topic.

- in lines 24-30, 32-38 it is necessary to add links to literature sources.

- Figure 3 is made very carelessly: some of the information is unreadable, there are also typos (in vitro should be in italics)

- typos in the title of Figure 3 also need to be corrected: in vitro should be in italics; 25 nm should be separate

- “Nanofingerprint” must be brought into a single pattern throughout the text (somewhere Nanofingerprint, and somewhere NanoFingerprint), this remark also applies to the designations Test 1, Test 3, etc. somewhere it is written together, and somewhere separately

- the size of nanoparticles is written without italics and separately

- Figure 4 contains the same errors as Figure 3

- it is not clear from the text of the article whether there are analogues of the ATENA website, this information must be added to section 4

- the “Discussion” section is more similar to the “Conclusion”, which is missing in this article; it is necessary to add an expanded “Discussion” section, with an analysis of the data obtained, comparison with experimental developments (if any), otherwise the reliability of the work performed can be questioned.

After changes have been made, the article may be reviewed again.

Comments on the Quality of English Language

Minor editing of English language required

Author Response

The article is devoted to the development of technologies that allow quantitative analysis of the structure, as well as the selection of areas of interest, to ensure effective analysis and processing of model nanostructures, as well as the ATENA web server for creating structural nanofingerprints. Today, the creation and improvement of tools that allow qualitative and quantitative analysis of various nanostructures is a very relevant area, especially with the use of digital fingerprints or nanofingerprints, as in this case. The article may be quite interesting for specialists working in the field of theoretical or experimental chemistry, however, I have several comments:

- the introduction is written too briefly, it lacks an analysis of works published on this topic or existing programs or services for creating models of nanostructures.

We have added a whole paragraph presenting 5 classical QSAR models for nanocompund metal-oxide toxicity prediction. The ones that the 3D structure was possible to be generated are later compared in the experimental section. Note these models have been implemented in ATENA, thus, currently, they can be easely used.

- in the introduction it is necessary to add information about the developments or scientific achievements previously obtained by the authors on this topic.

The following paragraph has been added at the end of the introduction “NanoFingerprints were  initially presented in a congress [14] without detailing examples, generation algorithms, their applicability and neither the web server. The aim of this paper is to explain to users (chemical labs or researches) this new model and how to use the web server.”

- in lines 24-30, 32-38 it is necessary to add links to literature sources.

We have added 5 QSAR examples (explained later) at the end of lines 24-30. And an example of using the structure in prediction at the end of lines 32-38. Thank you.

- Figure 3 is made very carelessly: some of the information is unreadable, there are also typos (in vitro should be in italics)

We have done the proper corrections. Thanks.

- typos in the title of Figure 3 also need to be corrected: in vitro should be in italics; 25 nm should be separate

We have done the proper corrections. Thanks.

- “Nanofingerprint” must be brought into a single pattern throughout the text (somewhere Nanofingerprint, and somewhere NanoFingerprint), this remark also applies to the designations Test 1, Test 3, etc. somewhere it is written together, and somewhere separately

Done, thanks.

- the size of nanoparticles is written without italics and separately

Done, thanks.

- Figure 4 contains the same errors as Figure 3

Done, thanks.

- it is not clear from the text of the article whether there are analogues of the ATENA website, this information must be added to section 4

To solve this issue, we have added the following paragraph in Section 4: “There are other web servers for the toxicity prediction of chemical compounds [21 ]. Nevertheless, these web servers propose general toxicity predictions and are not based on the structural information (XYZ file) and nor on nanocompounds.” Moreover, in the description of the three main functions (NanoFingerprints, Toxicity prediction, Subcomponent search), we have added whether exist some similar tool.

- the “Discussion” section is more similar to the “Conclusion”, which is missing in this article; it is necessary to add an expanded “Discussion” section, with an analysis of the data obtained, comparison with experimental developments (if any), otherwise the reliability of the work performed can be questioned.

The current version of the paper has these two sections. In the discussion one, we discuss the obtained results and in the conclusions section, the paper is concluded. Thank you for your interesting comment.

After changes have been made, the article may be reviewed again.

Thank you,

Round 2

Reviewer 2 Report

Comments and Suggestions for Authors

I thank the authors for their attentive attitude to the comments, however, some comments on the article are still present:

- line 49: TiO2 - typo (needs to be corrected throughout the text, including in the titles of the figures).

- why are some units of measurement in italics and some not (nm/nm3))? Should units of measurement be italicized at all? The same question applies to chemical formulas. Why put them in italics? If necessary, please provide a uniform template throughout the article.

- "xyz" is somewhere given as "xyz", and somewhere "XYZ"

- line 367: TiO2 typo (zero instead of oxygen)

Author Response

Thank you again for your time and effort. I've corrected all typos you have found.

Thanks again.